# Impact of a short version of the CONSORT checklist for peer reviewers to improve the reporting of randomised controlled trials published in biomedical journals: study protocol for a randomised controlled trial

Benjamin Speich [1,2] Sara Schroter [3] Matthias Briel,[2,4] David Moher [5,6] Iratxe Puebla [7] Alejandra Clark,[7] Michael Maia Schlüssel [1,8] Philippe Ravaud,[9,10] Isabelle Boutron,[9,10] Sally Hopewell[1]

**Correspondence to**
Dr Benjamin Speich;
benjamin.speich@ndorms.ox.ac.uk

## ABSTRACT

**Introduction** Transparent and accurate reporting is essential for readers to adequately interpret the results of a study. Journals can play a vital role in improving the reporting of published randomised controlled trials (RCTs). We describe an RCT to evaluate our hypothesis that asking peer reviewers to check whether the most important and poorly reported CONsolidated Standards of Reporting Trials (CONSORT) items are adequately reported will result in higher adherence to CONSORT guidelines in published RCTs.

**Methods and analysis** Manuscripts presenting the primary results of RCTs submitted to participating journals will be randomised to either the intervention group (peer reviewers will receive a reminder and short explanation of the 10 most important and poorly reported CONSORT items; they will be asked to check if these items are reported in the submitted manuscript) or a control group (usual journal practice). The primary outcome will be the mean proportion of the 10 items that are adequately reported in the published articles. Peer reviewers and manuscript authors will not be informed of the study hypothesis, design or intervention. Outcomes will be assessed in duplicate from published articles by two data extractors (at least one blinded to the intervention). We will enrol eligible manuscripts until a minimum of 83 articles per group (166 in total) are published.

**Ethics and dissemination** This pragmatic RCT was approved by the Medical Sciences Interdivisional Research Ethics Committee of the University of Oxford (R62779/RE001). If this intervention is effective, it could be implemented by all medical journals without requiring large additional resources at journal level. Findings will be disseminated through presentations in relevant conferences and peer-reviewed publications. This trial is registered on the Open Science Framework (https://osf.io/c4hn8).

## Strengths and limitations of this study

► Pragmatic randomised controlled trial (RCT) with individual randomisation of real manuscripts describing RCTs submitted to a variety of journals.
► Main outcomes will be assessed from publicly available sources (ie, published articles).
► If this simple intervention is effective, it could be implemented by journals without requiring large additional resources at journal level.
► The intervention could not be included within the email from journal with the link to the manuscript for review, risking peer reviewers will potentially ignore the separate email containing the CONsolidated Standards of Reporting Trials reminder.

## INTRODUCTION
### Background and rationale

There is substantial agreement that well conducted and reported randomised controlled trials (RCTs) generate the most trustworthy evidence when evaluating newly developed or existing clinical interventions.[1–3] For clinicians, scientists and decision makers, published articles are often the only way to know how a study was conducted. In order to judge the internal and external validity of RCTs, it is crucial that these articles present transparent, accurate and unbiased information about the methods and conduct of the RCT.

To improve the quality and transparency of clinical and epidemiological research, the EQUATOR (Enhancing the Quality and Transparency of Research) Network was founded in 2006 and officially launched in 2008.[4–10] This international network, which

assists in the development of reporting guidelines and actively promotes their use, consists of methodologists, epidemiologists, reporting guideline developers, statisticians, clinicians and journal editors.

The CONsolidated Standards of Reporting Trials (CONSORT) is perhaps the most prominent reporting guideline, designed to help improve the transparency and quality of reporting of RCTs.[11–13] It guides authors, peer reviewers and journal editors on the minimum information to be included in published reports of RCTs to facilitate critical judgement and interpretation of results and consists of 25 items and a flow diagram. The last update of the CONSORT Statement was published simultaneously in 10 leading medical journals in 2010[13] and currently CONSORT is endorsed by over 600 journals worldwide.[14]

Despite some improvement in reporting following the endorsement of the CONSORT statement, there remain major reporting deficiencies in published RCTs.[3 15–21] For example, a study of 1122 RCTs indexed in PubMed in December 2012 found that many did not define the primary outcome (31%), state the sample size calculation (45%) or explain the method of allocation concealment (50%).[22] This lack of transparency is a major limiting factor for readers who assess an article in order to find the answer to a specific question; it is also a major problem for scientists who perform systematic reviews and meta-analyses.

### Evidence to date

Journals can play a vital role in improving the reporting of published RCTs. For example, a survey of journals' 'Instructions to Authors' in 2014 found that 63% (106 of 168) of biomedical journals mentioned CONSORT[23]; however of those journals only 38 (36%) required a completed CONSORT checklist on submission. Such implementation indicates some improvement over time compared with an assessment in 2007 when only 17 of 62 (27%) journals requested the CONSORT checklist on submission.[24] A study using interrupted time series analysis and assessing if the CONSORT checklist for reporting abstracts of RCTs had an effect on reporting quality found that results were better reported in journals which had an active editorial policy to implement the checklist.[25]

A scoping review conducted in 2017 by Blanco and colleagues summarised different interventions aimed at improving adherence to reporting guidelines.[26] They identified a number of different interventions, some of which had been evaluated at journals. However, all the interventions, except requesting submission of checklists from authors, required additional resources from the journal (eg, internal peer review by editorial assistants or an additional peer-reviewer round conducted by a senior statistician using appropriate reporting guidelines).[27–29] Therefore, it is unlikely that these interventions will be implemented in the majority of journals, especially smaller journals with limited resources. Another study found that providing authors with a web-based CONSORT tool, which combined different CONSORT extensions and provided authors with a customised checklist, did not improve reporting when used at the manuscript revision stage.[30] However, a study examining 'the nature and extent of changes made to manuscripts after peer review, in relation to the reporting of methodological aspects of RCTs' and 'the type of changes requested by peer reviewers' found that peer review did lead to some improvement in reporting.[27]

The role of peer reviewers and expectations of them is varied.[31] While CONSORT checklists are sometimes available for peer reviewers to check, they are not typically instructed to assess this information as part of their review and there have been no studies evaluating the effect of asking them to do this. We plan to evaluate the impact of giving peer reviewers a short version of the CONSORT checklist together with a brief explanation of the items and asking them to check if they are adequately reported.

## METHODS AND ANALYSIS
### Objective

The objective of this study is to evaluate the impact of giving peer reviewers, during the standard peer review process, a short version of the CONSORT checklist (C-short) together with a brief explanation of the items and asking them to check if they are adequately reported in the manuscript.

### Study design

This study is a multicentre superiority RCT with submitted manuscripts as the unit of randomisation (figure 1; allocation ratio 1:1). This study protocol was written in adherence to the SPIRIT guidelines (online supplementary file).[32]

### Study setting and eligibility criteria

The population will be defined on two levels: included journals and included manuscripts.

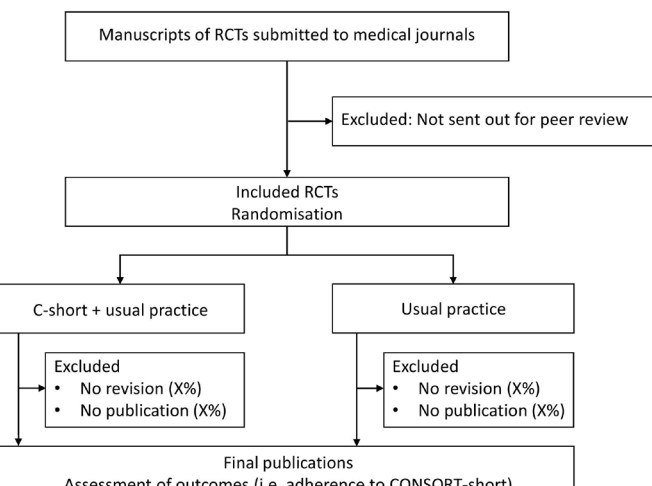

**Figure 1** Study flow chart. CONSORT, CONsolidated Standards of Reporting Trials; SPIRIT: Standard Protocol Items: Recommendations for Interventional Trials.

## Inclusion criteria for journals

Included journals must: (i) endorse the CONSORT statement by mentioning it in the journals' Instruction to Authors; (ii) have published primary results of at least five RCTs in 2017 (identified using a PubMed search). To be efficient, we plan to contact (via email) the editors of eligible journals from specific publishers (eg, BMJ Publishing Group; Public Library of Science (PLOS)) instead of separate journals. A description of the requirements for participation and a short summary information sheet will be included as part of the email invitation sent to journal editors. If a journal is eligible, and the editor agrees to take part, the editor will need to provide access to their editorial system (eg, ScholarOne, Editorial Manager) to enable the external researcher (BS) to screen and randomise eligible manuscripts. In cases where this is not possible, we will explore with individual journals if it would be possible to grant limited access (eg, only rights to screen studies) or to handle the different steps without access to the editorial system (eg, screening through automated reports; intervention provided by a journal staff member) and that the emails for the intervention would be sent by a member of the editorial team.

## Inclusion criteria for manuscripts

► All new manuscript submissions reporting the primary results of RCTs, which the journal editor has decided to send out for external peer review. Since the 10 chosen CONSORT checklist items (C-short) are applicable to different study designs, we will include all manuscripts reporting the primary results of RCTs regardless of study design (eg, parallel group trial, cluster trial, superiority trial, non-inferiority/equivalence trials).

## Exclusion criteria for manuscripts

► Manuscripts clearly presenting secondary trial results, additional time points, economic analyses or any other analyses.
► Manuscripts which are clearly labelled as a pilot or feasibility study or animal studies.
► Manuscripts not sent for peer review.

Details of journal manuscript submission and peer review processes, including consent and potential confidentiality issues, will be discussed in detail with each journal by teleconference and/or face to face prior to the journal agreeing to take part to ensure that randomisation of manuscripts is feasible.

In participating journals, the external researcher (BS) will check at least twice a week (by screening automated submission lists) all research manuscripts that are sent out for external peer review. As soon as the first invited peer reviewer accepts the invitation to review, the manuscript will be randomised to the intervention or control arm (see 'Randomisation' for more details). It is possible that this process might be slightly different among various included journals (eg, that team members of a journal might be involved in the screening if limited or no access to the journal's editorial system is granted).

## Interventions

### Control group: usual practice

After accepting to review a manuscript, peer reviewers will receive the automated, journal specific standard email with general information as per each journal's usual practice (eg, where to access the manuscript, date the peer review report is due).

### Intervention group: C-short plus usual practice

After accepting to review a manuscript, peer reviewers will receive the automated, journal specific standard email with general information (identical to control group). In addition, peer reviewers will receive an additional email from the editorial office that includes a short version of the CONSORT checklist (C-short) together with a brief explanation of the items either as a table within the email or as an attachment based on the preferences and possibilities of the journal (table 1, online supplementary appendix 1). Peer reviewers will be asked to check whether the items in the C-short checklist are addressed in the manuscript and to request authors to include these items if they are not adequately reported. This second email (see online supplementary appendix 1), containing the C-short checklist together with a brief explanation, is not generated automatically within the existing journal editorial systems (eg, ScholarOne or Editorial Manager); it will be sent manually by a researcher (BS) from the journal's editorial system or by a member of the journal's staff. In both cases the email will appear to have come from the editorial office (not the researcher).

### Development of the C-short checklist and explanation of items

For the development of C-short we chose the 10 most important and poorly reported CONSORT items as identified by a group of CONSORT experts in a previous study conducted by Hopewell and colleagues.[30] The selection of the items was based on expert opinion and empirical evidence whenever available.[30] In addition, to enable peer reviewers to better understand the items, we added a short explanation for each of the 10 items. These short explanations were extracted and amended from the CONSORT explanation and elaboration paper[11] and from COBWEB which is an online writing aid tool.[33] The short explanation was discussed and adapted by the scientific committee.

## Outcomes

### Primary outcome

The primary outcome of this study will be the difference in the mean proportion of adequately reported C-short items in published articles between the two groups.

### Secondary outcomes

Secondary outcomes will include the following:
► Mean proportion of adequately reported C-short items in published articles considering each item separately.

**Table 1** The 10 most important and poorly reported consort items as defined by a group of experts on the CONSORT statement.[30] for better understanding key features were summarised within a short explanation (extracted from the CONSORT explanation and elaboration paper[11] as well as from the COBWEB tool)[33]

| Item | Section | CONSORT item | Short explanation |
|---|---|---|---|
| 1 | Outcomes (6a) | Completely defined pre-specified primary outcome measure, including how and when it was assessed | Is it clear (i) what the primary outcome is (usually the one used in the sample size calculation), (ii) how it was measured (if relevant; eg, which score used), (iii) at what time point and (iv) what the analysis metric was (eg, change from baseline, final value)? |
| 2 | Sample size (7a) | How sample size was determined | Is there a clear description of how the sample size was determined, including (i) the estimated outcomes in each group, (ii) the $\alpha$ (type I) error level, (iii) the statistical power (or the $\beta$ (type II) error level) and (iv) for continuous outcomes, the SD of the measurements? |
| 3 | Sequence generation (8a) | Method used to generate random allocation sequence | Does the description make it clear if the 'assigned intervention is determined by a chance process and cannot be predicted'? |
| 4 | Allocation concealment (9) | Mechanism used to implement random allocation sequence (such as sequentially numbered containers), describing any steps taken to conceal the sequence until interventions were assigned | Is it clear how the care provider enrolling participants was made ignorant of the next assignment in the sequence (different from blinding)? Possible methods can rely on centralised or 'third-party' assignment (ie, use of a central telephone randomisation system, automated assignment system, sealed containers). |
| 5 | Blinding (11a) | If done, who was blinded after assignment to interventions (eg, participants, care providers, those assessing outcomes) | Is it clear if (i) healthcare providers, (ii) patients and (iii) outcome assessors are blinded to the intervention? General terms such as 'double-blind' without further specifications should be avoided. |
| 6 | Outcomes and estimation (17a/b) | For the primary outcome, results for each group, and the estimated effect size and its precision (such as 95% CI) | Is the estimated effect size and its precision (such as SD or 95% CI) for each treatment arm reported? When the primary outcome is binary, both the relative effect (risk ratio, relative risk or OR) and the absolute effect (risk difference) should be reported with CI. |
| 7 | Harms (19) | All-important harms or unintended effects in each group | Is the number of affected persons in each group, the severity grade (if relevant) and the absolute risk (eg, frequency of incidence) reported? Are the number of serious, life threatening events and deaths reported? If no adverse event occurred this should be clearly stated. |
| 8 | Registration (23) | Registration number and name of trial registry | Is the registry and the registration number reported? If the trial was not registered, it should be explained why. |
| 9 | Protocol (24) | Where trial protocol can be accessed | Is it stated where the trial protocol can be assessed (eg, published, supplementary file, repository, directly from author, confidential and therefore not available)? |
| 10 | Funding (25) | Sources of funding and other support (such as supply of drugs) and role of funders | Are (i) the funding sources, and (ii) the role of the funder(s) described? |

COBWEB, Consort-based WEB tool; CONSORT, CONsolidated Standards of Reporting Trials.

► Difference in mean proportion of adequately reported C-short items in published articles considering each sub-item (see 'Assessment of outcomes') as a separate item.

► Time from assigning an editor to the first decision (as communicated to the author after the first round of peer review).

- ► Proportion of manuscripts rejected after the first round of peer review.
- ► Proportion of manuscripts that will be published in the journal under study.

### Additional outcomes
- ► Exploratory analysis of available peer reviewer comments (ie, any references to CONSORT).

For journals where peer reviewers' comments are subsequently published alongside the published article, we will examine the peer reviewers' comments for any reference to CONSORT and trial reporting. We will contact those journals which do not make peer reviewers' comments publicly available, to see if reviews could be provided for such analyses under the condition that only anonymised data will be published.

### Assessment of outcomes
The outcomes will be assessed independently by two (blinded or at least partially blinded; see 'blinding') outcome assessors with expertise in the design and reporting of clinical trials. Any disagreement will be resolved by consensus or if necessary by consulting a third assessor. To ensure consistency between reviewers, we will first pilot the data extraction form; any disparities in the interpretation will be discussed and the data extraction form will be modified accordingly.

Adequate reporting of items will be assessed in duplicate from published full-text publications following the same instructions as provided by the CONSORT C-short checklist.[11] The following checklist items have, due to their complexity, sub-items which will be extracted separately. The sub-items are highlighted in the short explanation of the intervention (see table 1 and online supplementary appendix 1):
- ► Outcomes (item 6a): (i) define primary outcome, (ii) how it was measured, (iii) at what time point and (iv) the analysis metric (eg, change from baseline, final value).
- ► Sample size (item 7a): (i) the estimated outcomes in each group, (ii) the α (type I) error level, (iii) the statistical power (or the β (type II) error level), (iv) for continuous outcomes, the SD of the measurements.
- ► Blinding (item 11a): Is the blinding status clear for the following persons: (i) healthcare provider, (ii) patients and (iii) outcome assessors.
- ► Funding (item 25): (i) the funding source, and (ii) the role of funder in the design, conduct, analysis and reporting.

All items will be judged as either 'yes' meaning adequately reported, 'no' meaning not adequately reported or not reported at all, or 'NA' meaning that this sub-item is not applicable for this RCT. Items with different sub-items will only be judged as adequately reported if all relevant sub-items were adequately reported.

The outcomes 'time from assigning an editor to the first decision', 'proportion of manuscripts rejected after the first round of peer-review' and 'proportion of manuscripts that will be published in the journal under study' will be extracted directly from the journal's editorial system or provided by the journal.

### Participant timeline
The overview of the study schedule, including enrolment, intervention and assessments, is presented in table 2.

### Sample size
For the sample size calculation, we hypothesised in a first scenario (table 3) that the intervention C-short will result in a 25% relative increase in adequate reporting compared with the control (meaning that 70% of items will be adequately reported in the intervention group and 56% in the control group). This is based on a proportion of adequate reporting of 0.56 for the 10 most important and poorly reported items found in the control group of a previous study (meaning that a mean of 56% of the 10 most important and poorly reported items were reported).[30] The SD in the same study was 0.23. However, we calculated our sample size to account for a slightly larger variability in our data (SD=0.25). To demonstrate a significant difference with a power of 90% and a type 1 error at 5%, a total of 136 published articles will be required in this scenario (68 per treatment arm; based on a two sided t-test).

Two authors of this protocol, working for *PLOS ONE* (IP and AC), one of the participating journals, pointed out that 3 out of the 10 assessed items (ie, item 'Registration', 'Protocol' and 'Funding') should always be implemented in submissions to their journal given their policy requirements for clinical trials. Assuming that this journal will recruit a high proportion of manuscripts, and that also other journals might update their templates, we increased the sample size in a second scenario, in which all these three items would have an overall adherence of 90% in the control arm (table 3). This would entail an overall baseline adherence with the 10 C-short items of 71%. Based on a two sided t-test, a sample size of 166 (83 per treatment arm) will have a power of 80% to find a 15% relative increase (71% adherence in control group; 82% adherence in intervention group; SD=0.25; a type 1 error at 5%).

Since the final sample size will be based on the number of articles published, rather than on the number of manuscripts randomised, eligible manuscripts will be randomised until 83 articles are published in each arm (resulting in no less than 166 articles), to avoid loss of power due to potential imbalance between arms. Recruitment will be stopped as soon as both arms reach the sample size of 83. After recruitment has stopped we will wait 3 months so that manuscripts, which are still in production, can be published. Manuscripts which are published after the 3-month period will be excluded.

### Randomisation and blinding
Manuscripts meeting the eligibility criteria and sent out for external peer review by the journals will be randomised

**Table 2** Study schedule

| | Enrolment | Allocation and intervention | Intervention | Post-intervention | |
|---|---|---|---|---|---|
| Time point | Studies which are sent out for peer review | After first peer reviewer accepts invitation | Whenever an additional peer reviewer accepts invitation | First decision by journal | Published manuscripts |
| Eligibility screen | X | | | | |
| Allocation | | X | | | |
| Intervention: | | | | | |
| C-short + usual care | | X | X | | |
| Usual care | | X | X | | |
| Assessment of trial characteristics: | | | | | |
| Funding source | | | | | X |
| Study centres (single centre or multicentre) | | | | | X |
| Sample size | | | | | X |
| Study design (eg, parallel arm, crossover) | | | | | X |
| Hypothesis (eg, superiority, non-inferiority) | | | | | X |
| Medical field | | | | | X |
| Intervention tested | | | | | X |
| Number of trial arms | | | | | X |
| Number of peer reviewers | | | | | X |
| Journal which published the manuscript | | | | | X |
| Number of journals requesting CONSORT adherence (submission of checklist mandatory) | | | | | X |
| Assessment of outcomes: | | | | | |
| Time from assigning an academic editor until the first decision | | | | X | |
| Proportion of manuscripts directly rejected after the first round of peer review | | | | X | |
| Proportion of manuscripts that will be published in the journal under study | | | | | X |
| Adherence to CONSORT items and sub-items | | | | | X |

CONSORT, CONsolidated Standards of Reporting Trials.

into one of the two groups (allocation 1:1). The randomisation list will be created by the Study-Randomizer system[34] using random block sizes between 2 and 8 and stratified by journal. As soon as the first peer reviewer accepts the invitation, the manuscript will be included and randomised to one of the two study arms. One of the investigators (BS) will log onto the Study-Randomizer system[34] and enter the study identification number (ID;

**Table 3** Assumptions for sample size calculations in two different scenarios

| Item | CONSORT item | Scenario 1. adequate reporting as published in WebCONSORT[30] | Scenario 2. adapted from scenario 1 |
|---|---|---|---|
| 1 | Outcomes (6a) | 77% (79 of 103) | 77% (79 of 103) |
| 2 | Sample size (7a) | 83% (85 of 103) | 83% (85 of 103) |
| 3 | Sequence generation (8a) | 76% (78 of 103) | 76% (78 of 103) |
| 4 | Allocation concealment (9) | 55% (57 of 103) | 55% (57 of 103) |
| 5 | Blinding (11a) | 35% (36 of 103) | 35% (36 of 103) |
| 6 | Outcomes and estimation (17a | 44% (45 of 103) | 44% (45 of 103) |
| 7 | Harms (19) | 71% (73 of 103) | 71% (73 of 103) |
| 8 | Registration (23) | 69% (71 of 103) | 90% |
| 9 | Protocol (24) | 19% (20 of 103) | 90% |
| 10 | Funding (25) | 34% (35 of 103) | 90% |
| **Overall** | | **56%** | **71%** |

CONSORT, CONsolidated Standards for Reporting Trials

provided by the journal), the study title and the journal the study was submitted to. Subsequently, all additional peer reviewers accepting the invitation to review the same manuscript will receive the same group assignment as the first peer reviewer.

Authors will be blinded to the intervention. Editors will not be actively informed about the randomisation (possible exception listed under 'Interventions'). To avoid potential bias, peer reviewers and manuscript authors will not be informed of the study hypothesis, design and intervention.

Outcomes will be assessed in duplicate (see 'Assessment of outcomes'). At least one outcome assessor will be blinded. Due to restricted resources the investigator conducting the randomisation (BS) might be involved in the data extraction from published manuscripts.

## Data analysis

All quantitative variables will be described using means and SD, or medians and interquartile ranges in case severe departures from a normal distribution are identified. Data distributions will be inspected visually (ie, by histograms) instead of performing formal statistical tests for normality. Categorical variables will be described using frequencies and percentages. For the primary and secondary outcomes, we will estimate the mean difference

between the two groups and report them with respective 95% CI. No interim analysis will be conducted.

## Populations of analysis

The main population for analysis will be all manuscripts randomised and accepted for publication in the participating journals. In contrast to RCTs conducted with patients, where losses to follow-up need to be carefully considered (eg, multiple imputation of missing data), we are only interested in the reporting adherence of RCTs that are published. As such, we will exclude randomised manuscripts that were not published from the main analysis. All outcomes will be calculated based on the main population. The secondary outcome 'Time to the first decision' will additionally be calculated considering all randomised manuscripts (including the ones which were not published). For all analyses a p-value of 0.05 (5% significance level) will be used to indicate statistical significance. Exact p-values will be presented up to three decimal places. We anticipate there will be no missing data in this study, neither at the individual C-short items, nor at the manuscript level. This is due to the study design, which will include only the randomised manuscripts that are accepted for publication. We will analyse if the rate of manuscripts rejected after the first round of peer review and if the proportion of manuscripts that will be published differentiate among the two study arms (both secondary results).

## Analysis of primary endpoint

The effect of the intervention will be estimated as the mean difference in the proportion of C-short items adequately reported between the study arms. If the data on the primary outcome are normally distributed, groups will be compared using an unpaired Student's t-test. If the data are not normally distributed, comparisons will be performed using a non-parametric equivalent test (ie, Wilcoxon-Mann-Whitney test).

## Analysis of secondary endpoints

To investigate the effect of the intervention on the secondary outcomes, mean differences with respective 95% CI will be reported. If normality is not observed for any of the continuous secondary outcomes, the same strategy adopted for the primary outcome (use of a non-parametric equivalent to the Student's t-test) will be used.

## Pre-specified subgroup analysis

No formal subgroup comparative analysis is planned for the primary or secondary outcomes. However, the effect of the intervention on the primary outcome within subgroups will be presented using forest plots to visually examine whether it may differ according to some variables such as: (i) journals that actively implement the CONSORT Statement (defined as requiring authors to submit a completed CONSORT checklist alongside their manuscript) versus journals that are not actively implementing the CONSORT statement, (ii) sample size of included RCTs (n<100 vs n≥100) and (iii) impact factor

(<5, 5.1–10;>10) as there is evidence that higher impact factor and higher sample size are associated with higher adherence to reporting guidelines.[35] Subgroup analysis at the journal level will only be conducted when sufficient journals are in each group so that no results of individual journals are revealed. All analyses will be exploratory, with the aim of supporting new hypothesis generation, rather than being conclusive.

## Data management and confidentiality

Outcomes from publications will be assessed and extracted in duplicate. Since this information is not confidential, we will use freely available online forms (eg, Google forms) for data extraction from published RCTs. Data entered will be validated for completeness.

Data from the journal's editorial system (eg, title of manuscript, first author, randomisation ID, journal, date when manuscript was assigned to an editor, date when the final decision was made, final editorial decision, number of peer reviewers who reviewed the manuscript, the peer review reports (if available)) will be extracted (by BS or a member of the journal's staff), anonymised and entered in password protected files which are saved on a server from the University of Oxford. Data will be managed and curated according to University of Oxford regulations, which includes regular back-up (on a daily basis) of the virtual drives where the data are stored. No auditing or data monitoring is planned (as outcomes are directly extracted from journal's editorial system or in duplicate from published RCTs).

The raw data extracted from the included published manuscripts can be made openly accessible in an anonymised way (ie, giving the included RCT a number instead of identifying them). Derived/aggregated data, including anonymised information generated from the journal's editorial system, will be stored and made available to the research community when the project ends (see also 'Publication policy and access to data'). Where appropriate, the researcher who has access to the journal's editorial system (BS) and anyone else who will see the identifiable data will sign a confidentially agreement with the participating journals, confirming that they will not share identifiable data with any other party. Publishers such as the British Medical Journal (BMJ) state in their Company Privacy Statement that reviews and manuscripts may be used for quality improvement purposes and that is the nature of this research. Furthermore, peer reviewers for all BMJ journals receive the following statement in their invitation letter 'We are constantly trying to find ways of improving the peer review system and have an ongoing programme of research. If you do not wish your review entered into a study please let us know by emailing […] as soon as possible.'

## Trial registration

This trial was denied registration on ClinicalTrials.gov as the study is not a clinical study that assesses a health outcome in human subjects. Instead we registered the trial on the Open Science Framework (https://osf.io/c4hn8). The first manuscript was randomised in July 2019. We expect that recruitment will be finished in summer 2021.

## Patient and public involvement

Given the specific study topic, the steering committee agreed that patient or public involvement is not needed for this study.

## DISCUSSION

RCTs are the current gold standard for evaluating any new intervention in evidence-based medicine. Unfortunately, not all RCTs are of high quality. In fact, there are several well-known shortcomings with respect to reporting.[3 15–20] It is important to note that adhering to the CONSORT Statement does not mean that the study is of high quality. However, reporting all items from the CONSORT checklist will enable readers to adequately judge the quality of RCTs.

In this RCT we will test if a simple intervention in the form of asking peer reviewers to check whether selected CONSORT items are adequately addressed will increase the proportion of reporting completeness in the published RCTs in the participating journals. A multi-centre parallel arm RCT with randomisation at the individual manuscript level was chosen instead of a cluster RCT because the risk of 'contamination' at journal level was judged as low as the intervention will be implemented by an external researcher (ie, BS) or a member of the journal staff (eg, personnel from editorial services). The likelihood of contamination due to peer reviewers being invited to assess several RCTs and therefore becoming exposed to both intervention arms was judged small and therefore we do not plan to adjust for clustering by journal. Originally we planned to implement the intervention within the original instruction to peer reviewer email which is sent out as soon as a peer reviewer accepts the invitation from the journal. However, as these emails are sent automatically by the journal's editorial system we would have needed to modify the software from each journal to make sure that only half of the manuscripts administered the intervention. After our first discussion with journal editors and journal staff, we realised that this approach is not feasible and therefore decided to implement the intervention in the form of a separate email. We intended to conduct this RCT in a pragmatic way so that results 'would also be relevant to […] people who decide whether to implement the intervention on the basis of its results'.[36] Hence we chose to assess outcomes from published articles and not from manuscripts after the first round of revisions. Ideally, the full impact of the intervention would also be measured including all versions of randomised manuscripts in the final statistical analysis. However, due to confidentiality issues and limited resources we will not be able to evaluate manuscript versions prior to publication.

A selection of CONSORT items was chosen instead of the entire CONSORT checklist as we did not want to put too high a burden on peer reviewers, which could increase the risk that peer reviewers ignore our reminder.

Should the proposed intervention be successful in improving the reporting quality of published RCTs, as measured by the adherence to CONSORT, the intervention could be implemented at the journal level without requiring a large amount of additional resources. In addition, very similar interventions for other article types (eg, systematic reviews, trial protocols) and corresponding guidelines (eg, PRISMA [Preferred Reporting Items for Systematic Reviews and Meta-Analysis], SPIRIT) could be easily implemented too.

### Ethics and dissemination

Ethical approval has been obtained from the Medical Sciences Interdivisional Research Ethics Committee of the University of Oxford (R62779/RE001). The original approved study protocol is available in online supplementary appendix 2. The WHO trial registration data set is available in online supplementary appendix 3.

The results from this study will be published in a peer-reviewed journal irrespective of the study results. Authorship of publications will be granted according to the criteria of the International Committee of Medical Journal Editors. We plan to make the anonymised data set, including the data from the published articles, available as a supplementary file of the main publication.

**Author affiliations**
¹Centre for Statistics in Medicine, Nuffield Department of Orthopaedics, Rheumatology and Musculoskeletal Sciences, University of Oxford, Oxford, UK
²Basel Institute for Clinical Epidemiology and Biostatistics, Department of Clinical Research, University Hospital Basel, University of Basel, Basel, Switzerland
³The BMJ, London, UK
⁴Department of Health Research Methods, Evidence, and Impact, McMaster University, Hamilton, Ontario, Canada
⁵Centre for Journalology, Clinical Epidemiology Program, Ottawa Hospital Research Institute, Ottawa, Ontario, Canada
⁶School of Epidemiology and Public Health, Faculty of Medicine, University of Ottawa, Ottawa, Ontario, Canada
⁷PLOS ONE, Public Library of Science, Cambridge, UK
⁸The EQUATOR Network, Oxford, UK
⁹Université de Paris, CRESS, Inserm, INRA, F75004, Paris, France
¹⁰Centre d'Épidémiologie clinique, Hôpital Hôtel Dieu, Assistance Publique des Hôpitaux de Paris, Paris, France

**Contributors** SH, BS, IB, MB, DM and PR had the study idea and designed the study. SS, IP and AC provided expertise to ensure implementation at the journal level was possible. MMS was responsible for statistical aspects, including the sample size calculation and the data analysis plan. BS and SH wrote the first draft of the study protocol. All authors critically revised the protocol and approved the final version.

**Funding** BS is supported by an Advanced Postdoc.Mobility grant from the Swiss National Science Foundation (P300PB_177933). DM is supported by a University Research Chair, University of Ottawa. MMS is funded by Cancer Research UK. The funders had no role in designing the study and will also have no role in conducting the study, or analysing and reporting study results.

**Competing interests, and roles of the steering committee** SS is employed by the British Medical Journal (BMJ). IP and AC are employed by the Public Library of Science. DM, SH and IB are members of the CONsolidated Standards for Reporting Trials (CONSORT) executive and authors of the CONSORT 2010 statement. DM and PR are members of the EQUATOR (Enhancing the Quality and Transparency of Research) network steering group. MMS is a meta-researcher and reporting guideline developer, enthusiast and disseminator; he may therefore overestimate the importance of this project. All authors have declared that no other competing interests exist: the principal investigator (BS) is responsible for the preparation and the revisions of the study protocol, organising meetings of the steering committee, recruiting and randomising eligible manuscripts as well as the publication of study reports. The steering committee (IB, MB, SH, DM, PR, BS, MMS and SS) is responsible for revising the protocol, defining and validating the additional short explanation for each CONSORT item, advising on study implementation and for publishing the results of this study. MMS is responsible for the sample size calculation and the statistical analyses.

**Patient and public involvement** Patients and/or the public were not involved in the design, or conduct, or reporting or dissemination plans of this research.

**Patient consent for publication** Not required.

**Provenance and peer review** Not commissioned; externally peer reviewed.

**ORCID iDs**
Benjamin Speich http://orcid.org/0000-0002-3301-8085
Sara Schroter http://orcid.org/0000-0002-8791-8564
David Moher http://orcid.org/0000-0003-2434-4206
Iratxe Puebla http://orcid.org/0000-0003-1258-0746
Michael Maia Schlüssel http://orcid.org/0000-0002-1711-9310

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
