## [Reviewer comments · BMJ Open]

ARTICLE DETAILS

TITLE (PROVISIONAL)	Impact of a short version of the CONSORT checklist for peer reviewers to improve the reporting of randomised controlled trials published in biomedical journals: study protocol for a randomised controlled trial
AUTHORS	Speich, Benjamin; Schroter, Sara; Briel, Matthias; Moher, David; Puebla, Iratxe; Clark, Alejandra; Maia Schlüssel, Michael; Ravaud, Philippe; Boutron, Isabelle; Hopewell, Sally

VERSION 1 - REVIEW

REVIEWER	Fang Hua Centre for Evidence-Based Stomatology, School & Hospital of Stomatology, Wuhan University, China; Division of Dentistry, The University of Manchester, UK.
REVIEW RETURNED	31-Oct-2019

GENERAL COMMENTS	Congratulations to the authors on initiating such an important project. The protocol is very well-written and I only have the following minor comments: 1. [Line 101] - It seems that the EQUATOR Network was established in 2008 rather than 2006.2. [Line 141] - Another study by Cobo and colleagues (PMID: 22108262) also indicated that additional peer review using reporting guidelines is beneficial, please consider introducing it.3. [Line 245] - Please consider the following additional secondary outcomes: 1) time from peer reviewers' acceptance of invitation to their submission of review comments; 2) time from authors' reception of the decision letter (e.g. minor revision) to their submission of the revised manuscript; 3) proportion of rejected manuscripts (the C-short list may have a psychological influence on reviewers by reminding them of inadequate reporting and thereby add to their tendency to reject).4. [Line 389] - Please consider the following additional subgroup analysis: 1) peer reviewers' background (e.g. clinical expert vs. methodologist; from English speaking country vs. other countries); 2) Open Access journals vs. non-OA journals; 3) use of CONSORT guidelines / extensions in the initially submitted manuscript (mentioned vs. not mentioned).
---

REVIEWER	Ana Kowark Medical faculty University Hospital RWTH Aachen, Germany
REVIEW RETURNED	19-Jan-2020

GENERAL COMMENTS	It was an honour for me to review this highly important, well-designed study protocol. A transparent and meticulous reporting of randomised controlled trials (RCT) is essential for the appraisal by the readers. The CONSORT Statement is an important guideline, which facilitates the reporting and appraisal of RCTs. The authors are presenting a study protocol for an RCT, analysing the ability to enhance the adherence to the CONSORT checklist, by providing a shortened form of the CONSORT checklist (C-short) to peer-reviewers and instructing them to assess, whether the items are reported in the reviewed manuscript. Journals, willing to participate in this study, will be asked to provide information on submitted RCT manuscripts. The manuscripts will be randomised 1:1 into an intervention and a control group after the first peer-reviewer has accepted the invitation to review the manuscript. All peer-reviewers of the manuscripts, which are allocated to the intervention group, will receive a pre-specified instruction email from the editorial office, to check 10 items of the CONSORT checklist. The study will be terminated after 83 manuscripts in each arm are published. Two assessors (one blinded) will analyse independently the published articles regarding the primary and secondary outcomes. The primary outcome is the difference in the mean proportion of all reported C-short items between the two groups. The secondary outcomes comprise among others the mean proportion of each reported C-short item and subitem, respectively. The authors hypothesise that the proposed intervention will improve the adherence to the CONSORT statement in published articles. If this could really be revealed by this RCT, it would be easy to introduce this intervention in all journals, without a large amount of additional recourses. This would have a great impact on generating more trustworthy evidence from published articles. I wish the authors good success with this important study, which can easily be translated also to other kinds of manuscripts, like e.g. meta-analyses with the adherence to the PRISMA guideline.
--

VERSION 1 – AUTHOR RESPONSE

Reviewer(s)' Comments to Author:

Reviewer: 1

Reviewer Name: Fang Hua

Institution and Country:

Centre for Evidence-Based Stomatology, School & Hospital of Stomatology, Wuhan University, China;
Division of Dentistry, The University of Manchester, UK.

Please state any competing interests or state 'None declared': None declared.

Please leave your comments for the authors below

Congratulations to the authors on initiating such an important project. The protocol is very well-written and I only have the following minor comments:

Reply: We are grateful for the encouraging lines from Dr. Hua and for the overall positive feedback.

1. [Line 101] - It seems that the EQUATOR Network was established in 2008 rather than 2006.

Reply: We agree that our previous statement was unclear. Therefore, we clarify now in the revised version that the EQUATOR network was established in 2006 and officially launched in 2008. We also added the reference "A history of the evolution of guidelines for reporting medical research: the long road to the EQUATOR Network" written by Doug Altman and Iveta Simera, which nicely describes the emergence of the EQUATOR network (see revised manuscript, lines 93-95).

2. [Line 141] - Another study by Cobo and colleagues (PMID: 22108262) also indicated that additional peer review using reporting guidelines is beneficial, please consider introducing it.

Reply: We thank the reviewer for suggesting this reference and have included it in the revised version of our manuscript (see revised manuscript, lines 131-133).

3. [Line 245] - Please consider the following additional secondary outcomes: 1) time from peer reviewers' acceptance of invitation to their submission of review comments; 2) time from authors' reception of the decision letter (e.g. minor revision) to their submission of the revised manuscript; 3) proportion of rejected manuscripts (the C-short list may have an psychological influence on reviewers by reminding them inadequate reporting and thereby add to their tendency to reject).

Reply: We thank the reviewer for these suggestions. In general we would like to keep the number of outcomes to a small number, focussing on the most important ones. Additionally, we would prefer to not introduce new outcomes to the original study protocol which received ethical approval by the Medical Sciences Interdivisional Research Ethics Committee of the University of Oxford (R62779/RE001) and is include in Appendix 2. This does not mean that there will be certainly no additional outcomes or analyses in the future publication of the main results. However, in case we do include new outcomes or analyses in a publication of the main results, these will be clearly labelled as post-hoc analyses. Furthermore, we believe that the essence of the three suggested outcomes are well covered by the following three, already pre specified, outcomes (see revised manuscript, lines 248-251):

- Time from assigning an editor to the first decision (as communicated to the author after the first round of peer-review).
- Proportion of manuscripts rejected after the first round of peer review.
- Proportion of manuscripts that will be published in the journal under study.

4. [Line 389] - Please consider the following additional subgroup analysis: 1) peer reviewers' background (e.g. clinical expert vs. methodologist; from English speaking country vs. other countries); 2) Open Access journals vs. non-OA journals; 3) use of CONSORT guidelines / extensions in the initially submitted manuscript (mentioned vs. not mentioned).

Reply: As outlined above, in general we would like to focus in the outcomes and analysis on the most important aspects as well as not introducing new analyses which are not included in the original study protocol (see also answer to point 3 from reviewer 1). With respect to the specific proposed analyses, we are not aware how we would be able to conduct a sub-group analysis by peer reviewers background as usually at least two (and often more) peer reviewers are assessing the same manuscript. Currently it seems that rather few journals are participating in this trial. We have therefore included a statement that sub-group analysis at journal level will only be conducted when sufficient journals are in each group (i.e. to make sure that individual journals are not revealed; see revised manuscript, lines 395-397).

Reviewer: 2

Reviewer Name: Ana Kowark

Institution and Country: Medical faculty University Hospital RWTH Aachen, Germany

Please state any competing interests or state 'None declared': None declared

Please leave your comments for the authors below

It was an honour for me to review this highly important, well-designed study protocol. A transparent and meticulous reporting of randomised controlled trials (RCT) is essential for the appraisal by the readers. The CONSORT Statement is an important guideline, which facilitates the reporting and appraisal of RCTs.

The authors are presenting a study protocol for an RCT, analysing the ability to enhance the adherence to the CONSORT checklist, by providing a shortened form of the CONSORT checklist (C-short) to peer-reviewers and instructing them to assess, whether the items are reported in the reviewed manuscript. Journals, willing to participate in this study, will be asked to provide information on submitted RCT manuscripts. The manuscripts will be randomised 1:1 into an intervention and a control group after the first peer-reviewer has accepted the invitation to review the manuscript. All peer-reviewers of the manuscripts, which are allocated to the intervention group, will receive a pre-specified instruction email from the editorial office, to check 10 items of the CONSORT checklist. The study will be terminated after 83 manuscripts in each arm are published.

Two assessors (one blinded) will analyse independently the published articles regarding the primary and secondary outcomes. The primary outcome is the difference in the mean proportion of all reported C-short items between the two groups. The secondary outcomes comprise among others the mean proportion of each reported C-short item and subitem, respectively.

The authors hypothesise that the proposed intervention will improve the adherence to the CONSORT statement in published articles. If this could really be revealed by this RCT, it would be easy to introduce this intervention in all journals, without a large amount of additional recourses. This would have a great impact on generating more trustworthy evidence from published articles.

I wish the authors good success with this important study, which can easily be translated also to other kinds of manuscripts, like e.g. meta-analyses with the adherence to the PRISMA guideline.

Reply: We thank Dr. Kowark for this positive feedback and the encouraging words.

VERSION 2 – REVIEW

REVIEWER	Fang Hua Centre for Evidence-Based Stomatology, School & Hospital of Stomatology, Wuhan University, China; Division of Dentistry, The University of Manchester, UK.
REVIEW RETURNED	07-Feb-2020

GENERAL COMMENTS	The authors have addressed all my suggestions. I have no further comments, and recommend accepting this well-written study protocol.
--

REVIEWER	Dr. Ana Kowark Medical faculty University Hospital RWTH Aachen, Germany
REVIEW RETURNED	06-Feb-2020

GENERAL COMMENTS	Thank you for your careful revision. I have no additional comments.
---